# Entropy Rectifying Guidance
# for Diffusion and Flow Models

**Tariq Berrada Ifriqi**[1,2]      **Adriana Romero-Soriano**[1,3,4,5]
**Michal Drozdzal**[1]      **Jakob Verbeek**[1]      **Karteek Alahari**[2]
[1] FAIR at Meta    [2] Univ. Grenoble Alpes, Inria, CNRS, Grenoble INP, LJK, France
[3] McGill University    [4] Mila, Quebec AI institute    [5] Canada CIFAR AI chair
`tariqberrada@meta.com`

## Abstract

Guidance techniques are commonly used in diffusion and flow models to improve image quality and input consistency for conditional generative tasks such as class-conditional and text-to-image generation. In particular, classifier-free guidance (CFG) is the most widely adopted guidance technique. It results, however, in trade-offs across quality, diversity and consistency: improving some at the expense of others. While recent work has shown that it is possible to disentangle these factors to some extent, such methods come with an overhead of requiring an additional (weaker) model, or require more forward passes per sampling step. In this paper, we propose *Entropy Rectifying Guidance* (ERG), a simple and effective guidance method based on inference-time changes in the attention mechanism of state-of-the-art diffusion transformer architectures, which allows for simultaneous improvements over image quality, diversity and prompt consistency. ERG is more general than CFG and similar guidance techniques, as it extends to unconditional sampling. We show that ERG results in significant improvements in various tasks, including text-to-image, class-conditional and unconditional image generation. We also show that ERG can be seamlessly combined with other recent guidance methods such as CADS and APG, further improving generation results.

## 1    Introduction

Diffusion (Sohl-Dickstein et al., 2015; Ho et al., 2020; Song et al., 2021; Dhariwal and Nichol, 2021) and flow models (Lipman et al., 2023; Ma et al., 2024; Esser et al., 2024) are state-of-the-art generative modeling tools for various modalities, ranging from images (Rombach et al., 2022; Podell et al., 2024; Chen et al., 2024; Esser et al., 2024), to audio (Wang et al., 2023; Levy et al., 2023), and video (Jin et al., 2025; Polyak et al., 2024). These models generate data by starting with a simple prior and iteratively removing noise – a process called "denoising". See Kingma and Gao (2023); Lipman et al. (2023) for details on diffusion and flow matching. These models can be conditioned on various inputs to control the generative process, e.g., in text-to-image models. Guidance techniques such as classifier guidance (Dhariwal and Nichol, 2021) and classifier-free guidance (Ho and Salimans, 2021) are commonly used to improve sample quality and consistency with input conditioning. In the sampling process, these techniques combine a conditional signal with an unconditional one, and control the strength of conditioning through a scaling parameter. Although such guidance techniques are a crucial component in achieving state-of-the-art results, they are also known to negatively affect the diversity of generated samples given a particular prompt (Sadat et al., 2024; Karras et al., 2024; Kynkäänniemi et al., 2024; Saharia et al., 2022). Moreover, too high guidance scales may lead to overly saturated images, affecting the quality of generated images (Saharia et al., 2022). An extensive analysis on the trade-offs of image generation quality, diversity, and consistency was presented by Astolfi et al. (2024). To mitigate these quality-diversity-consistency trade-offs, more advanced

39th Conference on Neural Information Processing Systems (NeurIPS 2025).

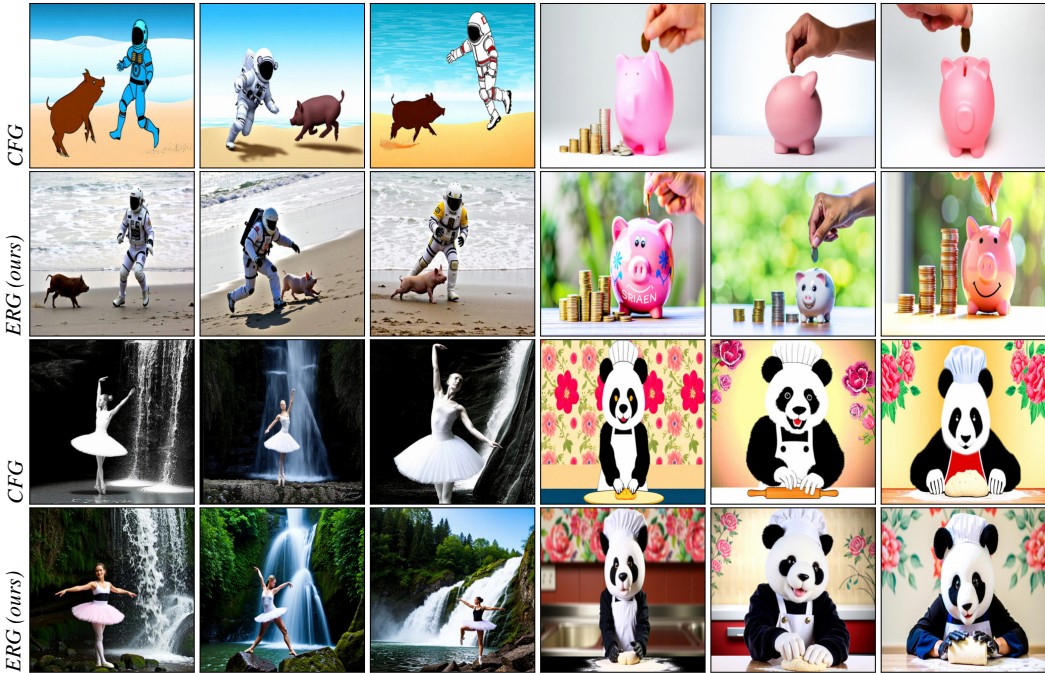

Figure 1: **Qualitative comparison of classifier-free guidance (CFG) and our Entropy Rectifying Guidance (ERG).** The images generated using ERG (second and fourth row) exhibit greater quality and diversity than standard CFG. Images are generated using 50 Euler steps; each column corresponds to a different random seed for the generations.

guidance techniques have recently been proposed, see e.g. Sadat et al. (2025); Karras et al. (2024). Most guidance techniques , however, require spending part of the training cycles on unconditional generation for the guidance to work, even if unconditional generation is not a goal in itself, and are not applicable to unconditional sampling (Sadat et al., 2025; Ho and Salimans, 2021). Others rely on a second model with weaker performance than the main model, thereby increasing memory requirements (Karras et al., 2024).

In our work, we build upon the work of Karras et al. (2024) and Hong (2024), and propose Entropy Rectifying Guidance (ERG): a simple and effective method to obtain *both* a *strong* and a *weak* predictive signal from a *single model* that leverages attention layers, where the model may be conditional or unconditional. In particular, our method uses the Hopfield energy formulation of attention (Ramsauer et al., 2021; Hong, 2024) and applies a temperature scaling to the softmax function of the attention layers in order to obtain the weak predictive signal. This scaling does not require any adaptations in model training, and may be applied to pre-trained denoising models, as well as their accompanying text encoders. Moreover, motivated by this energy interpretation of attention layers, we also consider iterative re-application of attention layers, and rescaling the residual attention update. Our experiments show that manipulating the attention layers in this manner results in simultaneous improvements in sample quality, diversity, and consistency, contrary to the trade-offs observed by Astolfi et al. (2024) for classifier-free guidance.

In summary, our contributions are the following:
1. We propose Entropy Rectifying Guidance (ERG), a guidance mechanism based on modifying the energy landscape of the attention layers.
2. Since our guidance mechanism does not require unconditional inference, it is directly applicable to any attention-based diffusion or flow model, including unconditional, class-conditional, and text-to-image models.
3. Experimentally, we find that ERG significantly improves image quality *and* diversity while retaining the same prompt consistency as the standard classifier-free guidance (+30 points in density, +4 points in VQAScore) on COCO at 512 resolution using our 1.9B text-to-image model.

# 2 Related work and background

In this section, we briefly review background material on diffusion and flow models, and related work on guidance techniques to improve sampling as well as the Hopfield energy formulation of attention.

## 2.1 Diffusion and flow matching models

Diffusion models (Sohl-Dickstein et al., 2015; Ho et al., 2020; Song et al., 2021) form a flexible class of generative models whose underlying principle is to map samples $\epsilon$ from a trivial unit Gaussian prior $p_0 = \mathcal{N}(0, \mathbf{I})$ to samples from a learned model $p_1$ of the data distribution. The forward process is defined as: $\mathbf{x}_t = \alpha_t \mathbf{x}_1 + \sigma_t \epsilon$ with $t \in [0, 1]$, where $\mathbf{x}_1 \sim p_1$, and $\alpha_t$ is a decreasing function of "time" $t$ while $\sigma_t$ an increasing function of $t$.

Flow matching methods (Lipman et al., 2023) assume that $\alpha_0 = \sigma_1 = 1$ and $\alpha_1 = \sigma_0 = 0$. Using these assumptions, during the reverse process $\mathbf{x}_t$ interpolates between $\epsilon$ at $t = 0$ and $\mathbf{x}_1$ at $t = 1$. In contrast, score-based diffusion models (Ho et al., 2020; Dhariwal and Nichol, 2021) set $\alpha_t$ and $\sigma_t$ implicitly through different formulations of stochastic differential equations (SDE) where $\mathcal{N}(0, \mathbf{I})$ is the equilibrium distribution. Additionally, they consider $t \in [0, T]$ with $T$ large enough so that $\mathbf{x}_T$ is approximately distributed as a unit Gaussian random variable.

## 2.2 Guidance mechanisms

**Classifier guidance.** To enable high quality conditional generation, Dhariwal and Nichol (2021) proposed to guide the sampling process by leveraging gradients from pre-trained auxiliary classifier $p(c|\mathbf{x})$ in each denoising step. They use the classifier to define the (scaled) joint score function as $\nabla_{\mathbf{x}_t} \log p(\mathbf{x}_t, c) = \nabla_{\mathbf{x}_t} \log p(\mathbf{x}_t) + w \nabla_{\mathbf{x}_t} \log p(c|\mathbf{x}_t)$, where $p(\mathbf{x}_t)$ is an unconditional data model, and $w$ is a scalar parameter regulating the strength of the classifier guidance. While classifier guidance allows to improve input consistency and image quality (at the expense of diversity), it requires an auxiliary classification model that is robust to inputs $\mathbf{x}_t$ with varying amounts of noise.

**Classifier-free guidance.** To avoid the need for an auxiliary noise-robust classifier, Ho and Salimans (2021) proposed classifier-free guidance (CFG). In this case, during the training process, two generative models are learned, one conditional $p(\mathbf{x}|c)$ and one unconditional $p(\mathbf{x}|\emptyset)$. In practice, the unconditional model is trained by dropping conditioning information with a small probability. The score function used for sampling is extrapolated towards the conditional prediction and away from the unconditional prediction $\nabla_{\mathbf{x}}^{\text{CFG}} \log p(\mathbf{x}|c) = w \nabla_{\mathbf{x}} \log p(\mathbf{x}|c) + (1 - w) \nabla_{\mathbf{x}} \log p(\mathbf{x}|\emptyset)$. While CFG improves image quality and input consistency with respect to classifier guidance, it tends to come at the cost of a reduction in diversity (Astolfi et al., 2024; Ho and Salimans, 2021; Sadat et al., 2024). Moreover, CFG often leads to generation artifacts, such as over-saturation as the guidance scale $w$ grows (Sadat et al., 2025).

**Advanced guidance techniques.** Several improved variants of classifier-free guidance have been proposed recently. Hong et al. (2023) presented Self-Attention Guidance (SAG), a guidance mechanism based on feeding a modified intermediate sample $\mathbf{x}_t$ when performing inference for unconditional prediction. The modification consists in blurring $\mathbf{x}_t$ in regions that are most attended to by the model's self-attention. This method has been developed for the U-Net architecture, hence applying it to more recent diffusion transformer architectures requires a hyperparameter search to understand which attention layers should be used for this method.

Smoothed energy guidance (SEG) (Hong, 2024) contrasts conditional prediction with a "weaker" conditional prediction obtained by altering the attention's softmax energy with a Gaussian kernel applied to queries. This method is developed for U-Net-style architectures and the softmax alteration applies to the self-attention layers in the middle block of the U-Net. In the conditional case, SEG uses a linear combination of the conditional, unconditional, and energy-smoothed unconditional prediction, whereas in our approach we only use the conditional and smoothed conditional term. Thus, SEG requires an additional function evaluation with respect to ERG for conditional inference. For unconditional inference, both approaches require only two function evaluations. In a similar spirit, Ahn et al. (2024) propose a guidance method based on manipulating the attention mechanism, by replacing the attention matrix with an identity mapping inside the denoiser U-Net.

Karras et al. (2024) proposed AutoGuidance, a method that uses a smaller/weaker version of the same conditional model for classifier-free guidance, resulting in better diversity and image quality. Like ours, their approach can also be applied to unconditional sampling. However, their method requires access to an earlier checkpoint of the model or, for best results, training a separate model with lower capacity, as well as accessing two models when sampling, which may increase the memory footprint.

Rather than considering modifications of the unconditional model term, Sadat et al. (2024) proposed the "condition-annealed diffusion sampler" (CADS) to increase the diversity of generations while maintaining sample quality. This is achieved by adding Gaussian noise to the conditioning tokens during inference, using a piecewise-linear decreasing schedule on the noise amplitude. Sadat et al. (2025) propose "adaptive projected guidance" (APG), a variant of CFG that resolves the over-saturation problem by emphasizing guidance orthogonal to the conditional prediction, rescaling the guidance term, and introducing a negative momentum term. The latter two approaches can be easily combined with our approach, and we consider such combinations in our experiments.

Lastly, several works (Kynkäänniemi et al., 2024; Chung et al., 2024a; Pavasovic et al., 2025; Wang et al., 2024) explore non-constant weight schedules for CFG. Such methods are complementary to our work, as our method only operates at the architecture level by rectifying the attention updates. Although they therefore could be combined with our approach, we defer this to future work.

## 2.3 Hopfield energy formulation of attention

The Hopfield network (Hopfield, 1982) is a dense associative memory model that aims to associate an input with its most similar pattern. More specifically, it constructs an energy function to model an energy landscape that contains basins of attraction around the desired patterns. Modern Hopfield energy networks (Ramsauer et al., 2021) introduce a new family of energy functions that improve the storage capacity of the model and make it compatible with continuous embeddings. Specifically, the following energy functional matches a continuous $d$-dimensional state (query) pattern $\boldsymbol{\xi} \in \mathbb{R}^d$ with $N$ stored (key) patterns $\mathbf{X} = (\mathbf{x}_1, ..., \mathbf{x}_N) \in \mathbb{R}^{d \times N}$ as $E(\boldsymbol{\xi}; \mathbf{X}) = \frac{1}{2}\boldsymbol{\xi}^\top \boldsymbol{\xi} - \mathrm{LogSumExp}\left(\mathbf{X}^\top \boldsymbol{\xi}, \beta\right)$, where $\mathrm{LogSumExp}(\mathbf{x}, \beta) = \beta^{-1} \log\left(\sum_{i=1}^d \exp(x_i)\right)$, where $x_i$ are the elements of the vector $\mathbf{x}$, and $\beta$ a scalar hyperparameter defining the sharpness of the approximation of the maximum in the LogSumExp operation. Intuitively, the first term imposes a finite norm on the queries while the second term measures the alignment between the state patterns (queries) and stored patterns (keys). Using the Concave-Convex Procedure (CCCP) (Yuille and Rangarajan, 2003), an iterative update rule, which converges to the global minima of the energy, can be derived as $\boldsymbol{\xi}_{l+1} = \mathbf{X}\,\mathrm{SoftMax}(\beta \mathbf{X}^\top \boldsymbol{\xi}_l)$, where $\mathrm{SoftMax}(\mathbf{x}) = \exp(\mathbf{x} - \mathrm{LogSumExp}(\mathbf{x}, 1))$. Equivalently, each iterative update step can be seen as a gradient update in the negative direction of the energy $\nabla_{\boldsymbol{\xi}} E(\boldsymbol{\xi}; \mathbf{X}) = \boldsymbol{\xi} - \mathbf{X}\,\mathrm{SoftMax}(\beta \mathbf{X}^\top \boldsymbol{\xi})$. Taking a gradient descent step on this energy landscape with a step size $\gamma$ results in an update of the form $\boldsymbol{\xi}_{l+1} = \boldsymbol{\xi}_l - \gamma\left(\boldsymbol{\xi}_l - \mathbf{X}\,\mathrm{SoftMax}(\beta \mathbf{X}^\top \boldsymbol{\xi}_l)\right)$, and for $\gamma = 1$ we recover the CCCP update.

Ramsauer et al. (2021) show that the CCCP update is related to the standard attention operation as follows. Assuming there are $S$ state (query) patterns, and $N$ stored (key) patterns that can be mapped to keys, queries and values using linear transformations, he state pattern can be obtained through a concatenation: $\boldsymbol{\Xi} = [\boldsymbol{\xi}_1, \ldots, \boldsymbol{\xi}_S] \in \mathbb{R}^{d \times S}$. Then the attention map is given by $\boldsymbol{\Xi}_{l+1} = \mathbf{X}\,\mathrm{SoftMax}(\mathbf{X}^\top \boldsymbol{\Xi}_l)$. The keys, queries and values are obtained via linear projections as $\mathbf{K} = \mathbf{X}W_K^\top \in \mathbb{R}^{N \times d}$, $\mathbf{V} = \mathbf{X}W_V^\top \in \mathbb{R}^{N \times d}$ and $\mathbf{Q} = \boldsymbol{\Xi}_{l+1}W_Q^\top \in \mathbb{R}^{S \times d}$, respectively. By setting $\beta = \frac{1}{\sqrt{d}}$ and substituting this into the CCCP update rule, we obtain: $\mathbf{Q}_{l+1} = \mathrm{SoftMax}(\frac{\mathbf{Q}\mathbf{K}^\top}{\sqrt{d}})\mathbf{K} = \mathrm{SoftMax}\left(\frac{\mathbf{Q}\mathbf{K}^\top}{\sqrt{d}}\right)\mathbf{V}\left(W_K W_V^{-1}\right)^\top$. Hence, we observe the equation of attention up to a linear transformation, which provides an interpretation of the attention mechanism through the Hopfield energy lens. Such a formulation has been used in the Energy Based Cross-Attention method (EBCA) (Park et al., 2024) for adaptive context control in order to incorporate additional contexts into the generative process of a conditional diffusion model.

## 3 Guidance via entropy rectification

Similarly to previous approaches (Ho and Salimans, 2021; Hong, 2024; Karras et al., 2024), our approach is based on contrasting the (conditional) denoising estimate with a less powerful one. In particular, using the Hopfield energy interpretation of attention (Ramsauer et al., 2021), see

Section 2.3, we manipulate the energy landscape of the attention layer, by rectifying the entropy of the associations in the attention operation. The modified attention layers lead to lower quality predictions, which are used as the contrasting term for guidance. This approach does not require a second model, and can be used for both conditional and unconditional sampling. We refer to our approach as Entropy Rectifying Guidance (ERG).

## 3.1 Manipulating the energy landscape

Our method manipulates the energy landscapes by introducing two new test-time hyperparameters, $\alpha$ and $\tau$ in the energy function:

$$E(\boldsymbol{\xi}; \mathbf{X}) = \frac{1}{2}\boldsymbol{\xi}^{\top}\boldsymbol{\xi} - \alpha \cdot \text{LogSumExp}\left(\mathbf{X}^{\top}\boldsymbol{\xi}, \tau \cdot \beta\right), \tag{1}$$

where $\beta = \frac{1}{\sqrt{d}}$ is the default temperature of the attention attention update. The temperature rescaling parameter $\tau$ controls the sharpness of the softmax attention, and $\alpha$ the relative importance of the similarity between the state matching term compared to the norm of the state patterns. Temperature rescaling with $\tau$ is similar to the Gaussian blurring of the attention maps introduced in SEG (Hong, 2024), but allows for non-local smoothing of the attention maps. Additionally, the view of the attention layer as a CCCP update of the energy function, allows for consideration of different methods to minimize the energy landscape. For instance, taking $K$ gradient descent steps with step size $\gamma$, as illustrated in Algorithm 1.

---

**Algorithm 1** Entropy rectifying guidance

**Require:** $K \in \mathbb{N}$ number of gradient update steps.
**Require:** $\gamma > 0$ step size.
**Require:** $\alpha \in \mathbb{R}$ State pattern matching weight.
**Require:** $\tau > 0$ attention temperature.
**Require:** $\mathbf{K}, \mathbf{V} \in \mathbb{R}^{d \times N}$ keys and values.
**Require:** $\mathbf{Q} \in \mathbb{R}^{d \times S}$ queries.
  $k \leftarrow 0$
  **while** $k < K$ **do**
    $\mathbf{Q} \leftarrow \mathbf{Q} - \gamma \left(\mathbf{Q} - \alpha \cdot \text{softmax}\left(\tau \cdot \beta \mathbf{Q}\mathbf{K}^{\top}\right)\mathbf{V}\right)$
    $k \leftarrow k + 1$
  **end while**

---

Using different settings of the hyperparameters $\alpha, \gamma, \tau$ and $K$ allows us to manipulate the attention operation, and obtain noise estimates that deviate from the trained model. We expect these to be weaker estimates compared to those provided by the model, as it was trained with standard attention layers, i.e. with $\alpha = \gamma = \tau = K = 1$. When applying this rectification mechanism to the denoiser model, we refer to the method as image-ERG, or I-ERG for short. In particular, we apply it to the negative/unconditional prediction part of the classifier-free guidance, and only on certain layers of the network that we will identify in our experiments. Additionally, we impose a kickoff threshold $\kappa$ on the time steps in which guidance is applied.

## 3.2 Manipulating the energy of the text encoder

Besides the image denoising model, we can also manipulate the energy landscape of attention-based text encoders to obtain a weak version of the conditional embeddings. Let $c$ be the text prompt that is used as conditioning. The text tokens are obtained by feeding the prompt to the text encoder: $\text{Enc}(c) \in \mathbb{R}^{d_t}$, such as Llama (Grattafiori et al., 2024) or T5 (Chung et al., 2024b). Text tokens are then fed to the denoiser model through cross-attention layers.

To obtain a contrasting signal for guidance, we manipulate the energy landscape in the self-attention layers of the text encoder following Algorithm 1. More specifically, for every self-attention layer in the text encoder, we introduce a temperature hyperparameter in the softmax function. This enables us to change the strength by which keys and queries are being matched, resulting in a modified prompt embedding at the output. For the remainder of the manuscript, we refer to this method condition-ERG, or C-ERG for short. For simplicity reasons, for the text encoder, we only consider changing the temperature but not the step size $\gamma$, pattern matching weight $\alpha$, and number of update steps $K$.

## 3.3 Guidance update

When combining the energy modulations in the text-encoder and denoising model, we obtain our Entropy Rectifying Guidance (ERG) update:

$$\Delta_{\text{ERG}}(\mathbf{x}, c, t; \Theta_\xi) = w \cdot D(\mathbf{x}, \boldsymbol{\phi}_c, t) + (1-w) \cdot D^{\boldsymbol{\xi}}(\mathbf{x}, \boldsymbol{\phi}_c^{\tau}, t; \Theta_\xi), \tag{2}$$

where $D$ is the learned denoiser model with parameters $\Theta$ that are omitted for simplicity, $D^{\boldsymbol{\xi}}$ is the denoiser model where the attention layers have been replaced by the modified version presented in Equation (1), and $\Theta_\xi = \{\alpha, \gamma, \tau, K\}$ the set of hyperparameters introduced by ERG. We use $\boldsymbol{\phi}_c$ to denote the prompt embedding produced by the text encoder, while $\boldsymbol{\phi}_c^\tau$ denotes the embedding obtained with the modified attention layers.

Compared to standard CFG, the main differences are that (i) we replace the unconditional text embeddings with conditional embeddings obtained with the entropy-rectified attention mechanism (C-ERG), and (ii) the denoiser model for the negative/unconditional predictions also uses modified attention layers (I-ERG). Finally, the changes to the image denoiser are only applied after a certain point during sampling in order to not overly penalize the negative components of the ERG update at the start of sampling. For the text-encoder we apply the temperature scaling throughout the denoising process, so that at any stage we obtain a noise prediction that can be contrasted with the vanilla denoising signal.

Note that our approach can be combined with other approaches, e.g., CADS (Sadat et al., 2024) and APG (Sadat et al., 2025). We will explore such combinations in our experiments.

# 4 Experimental evaluation

## 4.1 Experimental setup

**Datasets and architectures.** We experiment with class-conditional and text-to-image models trained using rectified flow-matching (Lipman et al., 2023; Esser et al., 2024). We use a face-blurred version of ImageNet (Deng et al., 2009) to train class-conditional models at 256 and 512 resolution based on the XL/2 variant of the DiT architecture (Peebles and Xie, 2023), which is composed of 28 attention blocks with hidden dimension of 1152, resulting in 790M parameters. For the text-to-image model, we use an architecture similar to MMDiT (Esser et al., 2024), and train a 512 resolution model on a mix of a proprietary dataset of 320M text-image pairs and YFCC100M (Thomee et al., 2016), where all faces in YFCC100M have been blurred. Similar to MMDiT (Esser et al., 2024), the model uses a mix of different text encoders: Llama3-8B (Grattafiori et al., 2024) and Flan-T5-XL (Chung et al., 2024b). During training, each of the text encoders is disabled with a probability of $\sqrt{0.1}$, so that the probability of both encoders being disabled is around 10%. We enable both text encoders during inference time for text-to-image generation, and disable both text-encoders for unconditional image generation experiments. The architecture of the model is made of 38 blocks with a hidden dimension of $1,536$, resulting in approx 1.9B parameters. For both datasets, we recaption the images using both Florence-2 Large (Xiao et al., 2024) to obtain medium-length captions and PaliGemma-3B (Beyer et al., 2024) for shorter COCO-style captions. Additional techniques to improve training efficiency of this model, such as conditioning mechanisms and pre-training strategies, were adopted from Berrada et al. (2024). For the class-conditional model we use the asymmetric autoencoder of Zhu et al. (2023), while for the larger text-to-image model we use the SD3 autoencoder (Esser et al., 2024).

**Metrics.** We consider metrics for quality, diversity, and consistency (Astolfi et al., 2024). We measure *sample quality* with FID (Heusel et al., 2017) and density (Naeem et al., 2020); *sample diversity* is measured with coverage (Naeem et al., 2020) and FID; and *prompt consistency* is measured with CLIPScore (Hessel et al., 2021) and VQAScore (Lin et al., 2024). For evaluation of text-to-image and unconditional generation, we use the 40k COCO'14 validation image-caption pairs. For the class-conditional models, we sample 50 images for each of the 1,000 ImageNet classes and use the ImageNet validation set as a reference. All evaluated models are sampled using the Euler method with 50 sampling steps. We use the EvalGIM (Hall et al., 2024) library for all evaluations.

**Baselines.** In addition to the standard classifier-free guidance, we compare our method to several recent state-of-the-art guidance techniques: Condition-Annealed Diffusion Sampler (CADS) (Sadat et al., 2024), Adaptive Projected Guidance (APG) (Sadat et al., 2025), Smooth Energy Guidance (SEG) (Hong, 2024), and Auto-Guidance (Karras et al., 2024). For APG, we follow the recommendations from the paper and set $\gamma_{\text{APG}} = -0.5$, $\eta_{\text{APG}} = 0.0$, $r_{\text{APG}} = 5.0$. For CADS, we perform a grid search over $\tau_1^{\text{CADS}} \in [0.6, 0.8]$, $\tau_2^{\text{CADS}} \in [0.8, 1.0]$, $s^{\text{CADS}} \in [0.25, 1.0]$, $\psi^{\text{CADS}} = 1.0$. Since SAG, PAG and SEG were developed specifically for the U-Net architecture, we adapt these method for diffusion transformers (Peebles and Xie, 2023) by applying the method in the attention layers

Table 1: **Comparison of ERG with other guidance approaches for text-to-image generation.** We compare ERG to other state-of-the-art guidance approaches and mark the best result in each column in bold in the top part of the table. In the bottom part of the table we evaluate combinations of ERG with APG and CADS, and bold results when they surpass the results in the upper part of the table.

| Guidance \ Metric | FID ($\downarrow$) | Density ($\uparrow$) | Coverage ($\uparrow$) | CLIPScore ($\uparrow$) | VQAScore ($\uparrow$) | NFE ($\downarrow$) |
|---|---|---|---|---|---|---|
| CFG (Ho and Salimans, 2021) | 12.81 | 98.24 | 71.12 | 26.45 | 70.15 | 2 |
| APG (Sadat et al., 2025) | 11.88 | 104.07 | 73.06 | 26.54 | 72.47 | 2 |
| CADS (Sadat et al., 2024) | 11.93 | 101.01 | 72.99 | 26.76 | 73.36 | 2 |
| PAG* (Ahn et al., 2024) | 12.75 | 107.21 | 72.20 | 26.80 | 73.32 | 2 |
| SAG* Hong et al. (2023) | **11.68** | 103.58 | 72.74 | 26.81 | 72.16 | 2 |
| SEG* (Hong, 2024) | 16.87 | 87.77 | 61.91 | **26.86** | 73.59 | 3 |
| AutoGuidance (Karras et al., 2024) | 16.62 | 87.02 | 62.75 | 26.59 | 73.53 | 2 |
| ERG (ours) | 13.62 | **120.25** | **73.21** | **26.86** | **73.96** | 2 |
| ERG (ours) + APG | **11.37** | 115.08 | **80.50** | 26.74 | 73.55 | 2 |
| ERG (ours) + CADS | 12.87 | **128.54** | 76.23 | 26.75 | 73.45 | 2 |

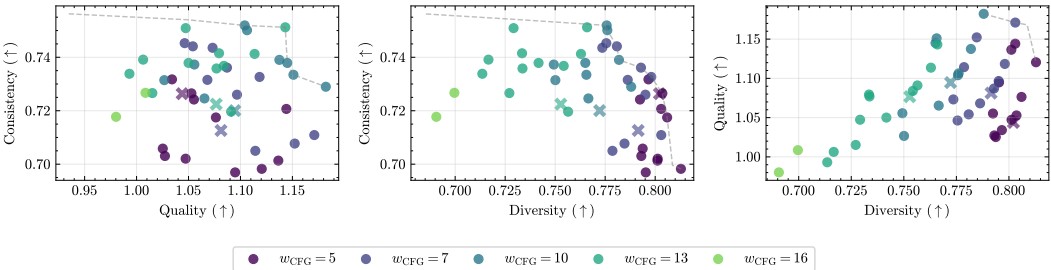

Figure 2: **Pareto fronts on consistency-diversity-quality for text-to-image generation.** Comparing ERG + APG (dots) with APG (crosses). We sweep over different guidance scales (each marked with a different color), and hyper-parameters $\alpha, \gamma, \tau$ for ERG. Dashed lines trace the Pareto fronts for each plot. We measure consistency with VQAScore, quality with density and diversity with coverage.

of the middle blocks of the transformer; we refer to these methods with an asterisk superscript. Additional details on the choice of layers are provided in Appendix D.4. For AutoGuidance, we follow the recommendations from the paper and use an earlier checkpoint of the same model, at approximately $1/16$-th of the training, as the weaker model. To ensure a fair comparison, we select the best performing guidance strength for baselines as well as our method using the rank-scoring algorithm detailed in Appendix A.2. Note that this is different from reporting the optimal score achieved for each metric, which might not correspond to any particular run because of inherent trade-offs between the different facets of the generations.

## 4.2 Main experimental results

Throughout our experiments, we modify attention layers in both the text encoder (C-ERG) and the image denoising model (I-ERG), unless specified otherwise.

**Text-to-image generation.** In Table 1, we compare ERG with recent state-of-the-art guidance mechanisms on the text-to-image generation task. ERG demonstrates excellent performance, outperforming baselines such as CFG, SAG, PAG and SEG* in most metrics. Specifically, ERG considerably boosts image quality as reported for Density (e.g., +22 points when compared to CFG). Additionally, ERG achieves the highest consistency scores: +0.4 points in CLIPScore and +3.8 in VQAScore when compared to standard CFG. Moreover, ERG combined with APG achieves the overall best diversity, measured by FID and Coverage, and also improves Density over prior methods. These results suggest that our ERG approach is successfully able to boost the generation quality of the model in all three facets of generation (quality, diversity, and consistency). Moreover, the number of function evaluations (NFE) required for our approach is only two per inference step, which is comparable to other methods except SEG which requires three.

Following Astolfi et al. (2024), we plot Pareto fronts for quality (measured by Density), diversity (Coverage), and consistency (VQAScore) metrics in Figure 2 when using ERG + APG. We find all

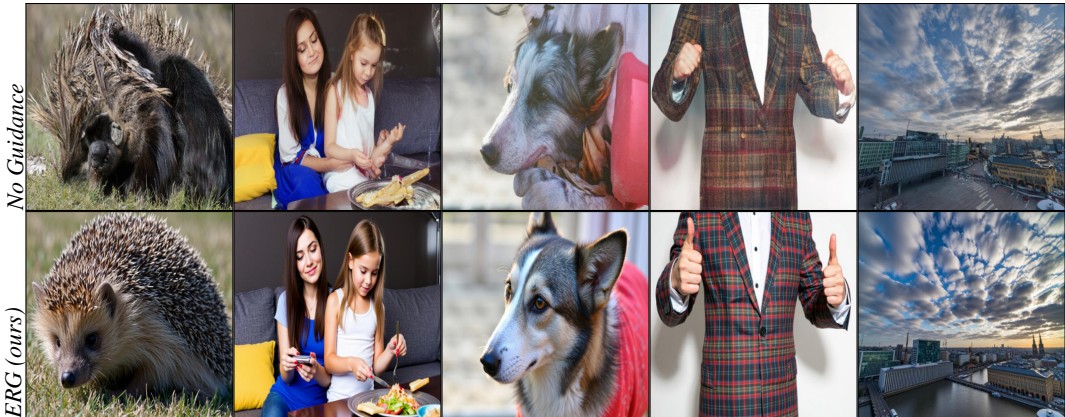

Figure 3: **Unconditional generation results.** Compared to not using guidance (top), our ERG generates more realistic and detailed images and more coherent structure (bottom). Images obtained from T2I model at 512 with empty prompt as input. Samples in each column use the same seed.

Table 2: **Unconditional generation.** Comparing ERG to other approaches compatible with unconditional sampling.

| | FID ($\downarrow$) | Density ($\uparrow$) | Coverage ($\uparrow$) |
|---|---|---|---|
| No guidance | 101.50 | 8.99 | 3.63 |
| SAG (Hong et al., 2023) | 39.25 | 46.22 | 30.70 |
| PAG* (Ahn et al., 2024) | 41.50 | 45.65 | 30.51 |
| SEG* (Hong, 2024) | 37.75 | 55.56 | 34.79 |
| AutoGuidance (Karras et al., 2024) | 39.50 | 48.26 | 34.71 |
| ERG (ours) | **36.25** | **55.84** | **51.59** |

Table 3: **Class-conditional generation.** Comparison of ERG with other guidance methods for models trained for 256 and 512 resolution.

| | Res. | FID ($\downarrow$) | Density ($\uparrow$) | Coverage ($\uparrow$) |
|---|---|---|---|---|
| CFG (Ho and Salimans, 2021) | | **3.67** | 127.03 | 85.81 |
| PAG* (Ahn et al., 2024) | | 5.31 | 111.94 | 81.04 |
| SAG (Hong et al., 2023) | 256 | 3.78 | 131.89 | 86.35 |
| SEG* (Hong, 2024) | | 6.15 | 132.22 | 84.51 |
| ERG (ours) | | **3.67** | **141.96** | **86.72** |
| CFG (Ho and Salimans, 2021) | | 5.65 | 146.97 | **86.70** |
| PAG* (Ahn et al., 2024) | | 4.65 | 134.49 | 86.50 |
| SAG (Hong et al., 2023) | 512 | 4.81 | 120.09 | 83.91 |
| SEG* (Hong, 2024) | | 6.59 | 160.11 | 81.85 |
| ERG (ours) | | **4.56** | **163.63** | 86.13 |

the points belonging to the Pareto fronts correspond to ERG + APG, which improves in all three facets of the generations w.r.t. APG, and provides significant boosts for quality and consistency. This can also be observed in the qualitative comparison between APG and ERG + APG in Figure 13 and Figure 14 in the supplementary material. Similarly, in Figure 1 the images sampled using ERG show better visual quality than those sampled with CFG.

**Unconditional generation.** In the unconditional model sampling experiment, we compare I-ERG with sampling without guidance and using applicable methods: AutoGuidance, SEG*, PAG* and SAG*. We provide quantitative evaluation results in Table 2, where we find that ERG outperforms all other methods, with significant boosts in FID, Density, and Coverage over other methods. The qualitative examples in Figure 3 clearly exhibit artifacts in terms of structural coherence in all the objects present in the generated images when not using guidance, which disappear when using ERG.

**Class-conditional generation.** For class-conditional generation, we observe similar trends as those seen for text-to-image and unconditional sampling in Table 3. In particular, at 256 resolution, we find improvements in Density and Coverage. FID remains similar to CFG but is better compared to all other methods tested. At 512 resolution, ERG is best across all metrics, except for coverage where ERG is slightly behind CFG and SEG*.

### 4.3 Analysis and ablations

**Text attention energy.** To isolate the effect of the temperature re-scaling in the text-encoder and image denoising components, we experiment with C-ERG and disable temperature rescaling in the image denoising model. In Figure 4, we vary the SoftMax temperature $\tau_c$ used in the text encoder, and consider generation performance for different guidance strengths $\omega$. For all metrics we find a somewhat symmetrical behavior around $\tau_c = 1$, which corresponds to an unguided prediction because in this case there is no difference with the normal conditional prediction.

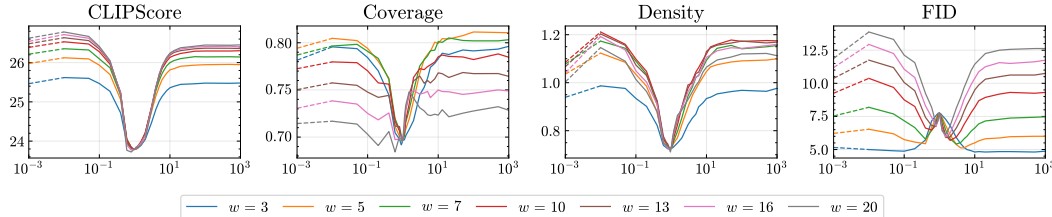

Figure 4: **Temperature rescaling in the conditioning for text-to-image generation (C-ERG).** We vary the text encoder's attention temperatures $\tau_c$. Each curve corresponds to a different guidance strength $w$. The left-most point on each curve represents the result for standard CFG.

Table 4: **Impact of the different components of ERG.** We accumulate different components of ERG, namely C-ERG, then I-ERG through denoiser entropy rectification and objective reweighting, finally all are merged into ERG.

| $\tau_c$ | $\tau_i$ | $\gamma$ | FID ($\downarrow$) | Density ($\uparrow$) | Coverage ($\uparrow$) | CLIP ($\uparrow$) | VQA ($\uparrow$) |
|---|---|---|---|---|---|---|---|
| ✗ | ✗ | ✗ | **12.81** | 98.24 | 71.12 | 26.45 | 70.15 |
| ✓ | ✗ | ✗ | 13.06 | 109.52 | 72.06 | 26.73 | 73.10 |
| ✓ | ✓ | ✗ | 13.62 | 120.25 | 73.21 | **26.86** | 73.96 |
| ✓ | ✓ | ✓ | 13.62 | **123.65** | **74.07** | 26.81 | **74.67** |

Table 5: **Multi-step gradient descent for optimizing the energy landscape.** we ablate different values for $K$ and $\gamma$.

| $K$ | 1 | 1 | 1 | 10 |
|---|---|---|---|---|
| $\gamma$ | 1 | 1.5 | 0.5 | 0.1 |
| FID ($\downarrow$) | 13.62 | 13.62 | 13.06 | **12.68** |
| Density ($\uparrow$) | 120.25 | **123.65** | 121.95 | 119.07 |
| Coverage ($\uparrow$) | 73.21 | **74.07** | 72.63 | 72.11 |
| CLIP ($\uparrow$) | **26.86** | 26.81 | 26.16 | 26.77 |
| VQA ($\uparrow$) | 73.96 | **74.67** | 73.01 | 73.95 |

The CLIPScore, Coverage and Density metrics generally improve when moving away from $\tau_c = 1$, making it either larger or smaller. For FID, the best values are obtained with low guidance scales and intermediate temperature values, the general trend shows improved FID using C-ERG compared to standard guidance. Compared with standard classifier-free guidance (left-hand side of the dashed lines), we find that all metrics can be improved for any guidance scale, provided with the right temperature. While $\tau_c < 1$ results in higher CLIPScore, $\tau_c > 1$ results in higher Coverage, indicating that tuning $\tau_c$ provides an easy way to control the diversity-consistency tradeoff.

**Combining the different parts.** In Table 4, we combine different parts of ERG and measure the effects on different facets of image generation. Our results show that all components show positive effects across all metrics, at the expense of a slight degradation in FID. Compared to the CFG baseline (first row), most of the improvement in CLIPScore and VQA are brought by the conditional entropy rectification ($\tau_c$, second row), while the improvements in Coverage and Density mostly come from rectifying the attention in the denoiser to obtain ($\tau_i$, third row). Finally, a further modest improvement in Density, Coverage and VQA is brought by the update step size ($\gamma$, fourth row).

**Multi-step gradient descent.** In Table 5 we consider the effect of varying number of updates $K$ for each attention operation along with the update step size. We find small variations in metrics with respect to the baseline $K = \gamma = 1$ with slight improvements when setting $\gamma = 1.5$, using multiple gradient descent steps did not induce significant gains. Therefore, we used $K = \gamma = 1$ in our default setup in our experiments, unless specified otherwise.

## 5 Conclusion

We presented *Entropy Rectifying Guidance (ERG)*, a novel guidance mechanism for sampling diffusion and flow models which significantly improves sample quality without sacrificing diversity and consistency performance. In particular, by manipulating the energy landscape of the attention layers in the diffusion transformer and the text encoder at inference time, ERG significantly boosts the performance of different models when studying their quality-consistency-diversity trade-offs and is applicable to different modalities such as text-to-image, class conditional and unconditional models. Furthermore, ERG outperforms recent state-of-the-art guidance methods such as CADS, APG, SEG and AutoGuidance, while requiring the same compute as standard CFG. ERG can be combined with approaches such as APG and CADS, which further improves results.

**Acknowledgements.** Karteek Alahari was supported in part by the Institute of Information & Communications Technology Planning & Evaluation (IITP) grant funded by the Korean Government (MSIT) (No. RS-2024-00457882, National AI Research Lab Project).

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
