# OpenReview forum: "Entropy Rectifying Guidance for Diffusion and Flow Models"
_NeurIPS.cc/2025/Conference — NeurIPS 2025 poster_

### Official Review · Reviewer_R39s · 2025-06-23

**Clarity:** 3
**Significance:** 3
**Originality:** 3
**Rating:** 5
**Confidence:** 3

**Summary:**

This paper proposes Entropy Rectifying Guidance (ERG) which acts as a inference time modification of attention. ERG claims to improve quality and diversity simultaneously. Furthermore the authors also show their method applied to the text encoder and diffusion models separately.

**Questions:**

- Why is ERG so much better in Coverage in the unconditional case while being only marginally better in FID and density?
- Why is there such a sharp change in measured performance in Fig 8?

**Ethical Concerns:**

["NO or VERY MINOR ethics concerns only"]

**Final Justification:**

After the rebuttal, the authors have answered all questions I had and explained any topic I was confused on in a precise manner. Therefore, i am raining clarity/quality to 3 and the final rating to Accept.

**Limitations:**

The authors discuss limitations.

**Paper Formatting Concerns:**

None.

**Quality:**

3

**Strengths And Weaknesses:**

Weaknesses:
- Not enough transparancy in terms of evaluation. Datasets should be stated more clearly.
- The authors should include ImageNet 256 x 256 using a pretrained DiT-XL/2 and cite the original metrics from the respective papers.
- Further, a comparison using sFID, InceptionScore, Recall and Precision would be good as this gives even more ground for fair comparisons.

Strengths:
- Overall well written and interesting concept.
- Metrics support their general claims even though i would like to see a bit more common metrics as means of comparison (see W2/W3)
- Very good performance in the unconditional generation which is interesting.

---

> ### Author Rebuttal · Authors · 2025-07-30
>
> We thank the reviewer for their valuable feedback.
> We are especially encouraged that they recognized the novelty of ERG as an inference-time attention modulation and appreciated its ability to improve both quality and diversity.
> We are glad they found the concept interesting and the paper well written.
> Their acknowledgment of ERG’s strong performance in the unconditional generation setting is very much appreciated.
>
> Below we address all the questions raised by the reviewer.
>
> **W1.** “*Not enough transparency in terms of evaluation. Datasets should be stated more clearly.*”
> We thank the reviewer for raising this point. We discuss the datasets and the evaluation setup in Section 4.1.
> We will make sure to update the experimental section to separate dataset setup and architecture details in order to improve clarity.
> If the reviewer points to specific information that they think is missing from the text or not clearly explained, we would be happy to address it during the discussion period and update it in the paper.
>
> **W2.** "*The authors should include ImageNet 256 x 256 using a pretrained DiT-XL/2 and cite the original metrics from the respective papers.*"
> As suggested, we evaluated with DiT-XL/2 using the open-source implementation, and show the results below. The positive influence of ERG is clearly demonstrated in these results.
>
> | Model \ Metric | FID | sFID | Precision | Recall | IS |
> | :---- | :---- | :---- | :----: | :---- | :---- |
> | DiT-XL/2-256 | 2.71 | 4.55 | 82.77 | 57.94 | 277.96 |
> | DiT-XL/2-256 \+ ERG |  2.36 | 4.32 | 86.90 | 60.03 | 295.03 |  |
>
> **W3.** "*A comparison using sFID, InceptionScore, Recall and Precision would be good as this gives even more ground for fair comparisons.*"
>
> We thank the reviewer for this suggestion. The extended list of metrics for our main results in Table 1 of the paper is presented below.
>
> | Method | CFG | APG | CADS | PAG | SAG | SEG | AutoGuidance | ERG | ERG+APG | ERG+CADS |
> | :---- | :---- | :---- | :---- | :---- | :---- | :---- | :----: | :---- | ----: | :---- |
> | Precision | 65.71 | 66.34 | 66.42 | 66.42 | 64.97 | 61.22 | 61.48 | 70.92 | 69.50 | **72.72** |
> | Recall | 43.95 | 45.15 | 45.75 | 43.94 | 47.98 | 36.88 | 34.60 | 41.43 | **50.25** | 38.89 |
> | sFID | 26.53 | 20.31 | 18.20 | 27.21 | 28.55 | 33.80 | 14.75 | 19.56 | **9.30** | 15.32 |
> | IS | 37.25 | 43.76 | 40.75 | 40.50 | 38.28 | 40.64 | 41.81 | 43.16 | **53.47** | 42.52 |
>
> For precision & recall, we observe similar trends to density and coverage with ERG+CADS achieving the best precision (72.72). ERG on its own achieves a precision of 70.92 compared to 65.71 for CFG.
> Recall is slightly lower when using ERG alone (41.43 vs 43.95 for CFG), but ERG+APG achieves the best recall of 50.25.
> For Inception Score, ERG improves over the CFG baseline (43.16 vs 37.25 for CFG) and achieves the best score when combined with APG (53.47).
> Similarly, ERG+APG results in significant sFID and Inception score improvement, with ERG alone on par with APG.
>
> **Q1.** “*Why is ERG so much better in Coverage in the unconditional case while being only marginally better in FID and density?*”
>
> ERG shows a stronger improvement in coverage in the unconditional case, while gains in FID and Density are more modest. This is largely due to our multi-metric optimization strategy: we tune our models to perform well across all metrics—coverage, FID, and density—rather than optimizing exclusively for one.
>
> Coverage captures the diversity of generated samples, whereas density reflects sample quality, and FID balances both. The significant boost in coverage specifically stems from applying a scheduled attention temperature, which allows ERG to gradually increase the influence of guidance over time. In contrast, SEG applies strong guidance from the very beginning, which tends to over-optimize for quality early in generation at the expense of diversity and consistency. This tradeoff is discussed in Section C.5 and illustrated in Figure 15 of the appendix.
>
> **Q2.** “*Why is there such a sharp change in measured performance in Fig 8?*”
>
> As discussed in Section C.4 of the appendix, we observe that attention layers exhibit distinct functional roles across the network, which can be broadly grouped into three stages: (1) encoding (layers 0–10, low entropy), (2) modeling (layers 10–25, high entropy), and (3) decoding (layers 25–38, low entropy). We find that modifying the rectification mechanism in either the encoding or decoding stages leads to sharp drops in performance due to the emergence of low-level artifacts in the generated outputs. This explains the abrupt change observed in Figure 8, and the same effect can be seen in Figure 16\.

---

> > ### Comment · Reviewer_R39s · 2025-08-01
> >
> > Thank you for the rebuttal.
> >
> > W2: The original DiT-XL/2 shows results of 2.27 in terms of FID. Why is your replication so much worse?
> >
> > The authors have addressed the rest of my concerns well and gave clear explanations.

---

> ### Author Response · Authors · 2025-08-03
> **FID mismatch issue for DiT.**
>
> Thank you for raising this point.
>
> After careful investigation we realized that for the DiT experiment, we reported FID using the autoguidance evaluation code while all other metrics (sFID, Precision, Recall, IS) were reported using the code from guided-diffusion. While both libraries use the same Inception network, they both provide a set of reference statistics for ImageNet which are different.
>
> While evaluating the DiT generations (at 256 resolution) with the autoguidance codebase, we were using the ImageNet512 reference statistics under the hood, resulting in an incorrect setup, which explains the higher FID numbers.
>
> We re-ran the experiment using the guided-diffusion library to match exactly the evaluation setup from the DiT paper for 10 different random seeds. The results of the rectified experiment are reported below.
>
> |  | IS | FID | sFID | Precision | Recall |
> | :---- | ----: | :---- | :---- | ----: | ----: |
> | mean (baseline) | 278.67 | 2.38 | 4.55 | 82.74 | 57.96 |
> | std (baseline) | 4.93 | 0.05 | 0.12 | 0.16 | 0.41 |
> | min (baseline) | 270.67 | 2.31 | 4.41 | 82.48 | 57.34 |
> | max (baseline) | 273.39 | 2.48 | 4.55 | 82.75 | 58.50 |
> | Mean (I-ERG) | 293.79 | 2.15 | 4.28 | 86.94 | 59.98 |
> | std (I-ERG) | 2.06 | 0.03 | 0.08 | 0.09 | 0.42 |
> | min (I-ERG) | 289.18 | 2.09 | 4.26 | 86.82 | 59.21 |
> | max (I-ERG) | 297.84 | 2.18 | 4.33 | 87.06 | 60.64 |
>
> For the baseline, we obtain an average FID of 2.38 which is still higher than the one reported in the paper but is plausible given the variance of the results.
>
> Compared to the baseline, I-ERG still achieves a better FID (2.15 vs. 2.38), and given the statistics, also yields smaller variance on all metrics other than recall, showing more stable performance while varying random seeds.
>
> Thank you once again for pointing out this inconsistency.

---

> > ### Comment · Reviewer_R39s · 2025-08-04
> >
> > Thank you for the clarification. Now all my concerns are resolved.

---

### Official Review · Reviewer_1okZ · 2025-06-30

**Clarity:** 4
**Significance:** 3
**Originality:** 3
**Rating:** 5
**Confidence:** 3

**Summary:**

This paper proposes a novel guidance method to improve sampling quality by updating the model via test-time adaptation. Specifically, during the sampling process, the authors reinterpret the attention mechanism in the self-attention layer as a Hopfield energy formulation and perform updates that descend the Hopfield energy landscape. This approach achieves state-of-the-art results across various evaluation metrics, including FID, recall, coverage, density, prompt fidelity, and VQA scores.

**Questions:**

Please see weaknesses.

**Ethical Concerns:**

["NO or VERY MINOR ethics concerns only"]

**Final Justification:**

The rebuttal provided clear and convincing explanations for my concerns about the hyperparameter τ and the slight FID underperformance. Given these resolutions and the paper’s overall strength, I am raising my recommendation from Borderline Accept to Accept.

**Limitations:**

Please see weaknesses.

**Paper Formatting Concerns:**

.

**Quality:**

4

**Strengths And Weaknesses:**

# Strengths
- The idea of updating attention layers at test time, inspired by hopfield energy is novel.
- The method is simple yet effective.
- The paper is well-written.
- The authors conduct thorough comparisons across many evaluation metrics (FID, recall, coverage, density, prompt fidelity, VQA score).
- Extensive experiments are conducted on various models and datasets.
- The method also performs well in unconditional sampling scenarios.

# Weaknesses & Questions
- In Figure 4, the hyperparameter τ, which is not derived from the Hopfield energy function, appears to be very influential. It would be beneficial to provide a theoretical explanation or motivation for its inclusion.
- Although the method outperforms others on most metrics, it slightly underperforms in FID compared to some baselines. Is there a justification or insight into why this might be the case?

# Minor Weaknesses
- Line 151: Typo. “he state patter” → should be corrected.

---

> ### Author Rebuttal · Authors · 2025-07-30
>
> We thank the reviewer for their thoughtful and encouraging feedback. They highlighted the novelty of adapting attention layers at test time using a Hopfield energy perspective, as well as the simplicity and effectiveness of the proposed method. We appreciate their recognition of the thoroughness of our experimental evaluation across a wide range of metrics, models, and datasets. Their acknowledgment of the method's strong performance in both conditional and unconditional sampling scenarios is also greatly valued.
>
> Below we address all the questions raised by the reviewer.
>
> **W1. Motivation for $\tau$** We thank the reviewer for highlighting the importance of the temperature hyperparameter $\tau$. While not part of the original Hopfield energy formulation, τ is a theoretically grounded extension that modulates the sharpness of the softmax in the *LogSumExp* term of the energy function (eq. (1) in the paper). Specifically, it enables control over the entropy of the attention distribution, allowing us to interpolate between diffuse and sharply peaked attention.
>
> Mathematically, introducing $\tau$ rescales the similarity scores as $\log \sum_i \exp(\tau \beta x_i^T \psi)$, corresponding to a softmax temperature adjustment.
>
> From a statistical mechanics perspective, $\tau$ acts as an inverse temperature in a *Boltzmann distribution*, balancing exploration (low $\tau$, high entropy) and exploitation (high $\tau$, low entropy) when matching attention keys and queries.
>
> Crucially, introducing $\tau$ does not break the concave-convex structure of the energy function, and the CCCP optimization remains valid and convergent. This allows us to embed $\tau$ directly into the energy landscape and perform test-time entropy manipulation in a principled manner.
>
> Unlike prior methods such as SEG, which apply local Gaussian smoothing post hoc, our approach integrates $\tau$ into the optimization dynamics via CCCP descent, enabling non-local and theoretically interpretable control over attention.
>
> We will clarify this motivation and its theoretical basis in the revised paper.
>
> **W2. Under performance in terms of FID**
> We thank the reviewer for pointing out the slightly higher FID values in some configurations. Our evaluation strategy is designed to prioritize balanced performance across fidelity, diversity, and consistency, rather than optimizing for FID alone.
>
> As shown by [1], FID-optimal checkpoints often lie far from the Pareto front when jointly considering realism, consistency, and diversity, and in models such as SDXL, optimizing for FID can significantly reduce realism (Section 4.1, Fig. 3).
> Dhariwal et al. [2] further show that lower guidance scales minimize FID and recall, while higher guidance scales improve perceptual sharpness and structure, measured by IS and precision even though they increase FID (Figure 3).
>
> In line with these findings, we adopted a multi-objective evaluation strategy. To directly address this question, we conducted an additional experiment optimizing hyperparameters for FID. We will include results (presented below) for completeness.
>
> | Method | CFG | ERG | APG | ERG+APG | PAG | SAG | SEG | CADS | ERG+CADS | AG |
> | :---- | :---- | :---- | :---- | :---- | :---- | :---- | :---- | :---- | :---- | :---- |
> | FID | 5.25 | **4.93** | 6.62 | 5.24 | 7.00 | 5.28 | 6.72 | 5.12 | 5.00 | 7.11 |
>
>
> **Minor weaknesses**
> Thanks for pointing out this typo, it will be corrected in future versions of the paper.
>
> [1] Consistency-diversity-realism Pareto fronts of conditional image generative models, Astolfi et al., NeurIPS'24.
>
> [2] Diffusion Models Beat {GAN}s on Image Synthesis, Dhariwal et al., NeurIPS'21.

---

> ### Comment · Reviewer_1okZ · 2025-08-03
>
> Thank you for the clarification. All my concerns have been resolved after reading the rebuttal.

---

### Official Review · Reviewer_Rztq · 2025-07-03

**Clarity:** 3
**Significance:** 3
**Originality:** 2
**Rating:** 5
**Confidence:** 4

**Summary:**

The paper presents Entropy Rectifying Guidance (ERG), an inference-time guidance method presented as an alternative to classifier-free guidance (CFG) for diffusion/flow-matching models. ERG operates by modifying the energy landscape of attention layers within a diffusion/flow model. Particularly, the softmax temperature in attention is modulated to produce a weaker conditional signal/prediction, which takes the place of the unconditional estimation in CFG. A particularly interesting outcome of ERG is that it can be used to improve unconditional sampling, unlike CFG. The authors provide a set of convincing qualitative and quantitative experiments to demonstrate that ERG improves quality, diversity, and consistency.

**Questions:**

Q1. In section 4.1, the authors mention that they train their own diffusion/flow models for each of the experiments. Can the existing method not be used as a drop-in replacement for CFG in pre-trained diffusion models?

- If training is necessary: why is it required? It seems to me that ERG can be used in pre-trained models, unless I missed something.
- If training is not necessary: why not provide some experiments with existing open-source pre-trained models?

Q2. Instead of converting some prior work on attention-based UNet architectures to DiT for comparison (SAG, PAG, SEG, noted in L263), it would have been better if the authors could have directly applied their method on the attention-based UNets and compared against the previous methods without modifying them. It will address W1.1. [Unless, ERG requires training a model from scratch (as I noted in Q1), in which case this comparison is not necessary.]

Q3. From my understanding, SEG uses an extra NFE simply because it wants to utilize the already available unconditional pretrained model. Can it not be used similar to ERG in the conditional case without an extra NFE?

- The following ablation would help make things clear: (a) disable temperature scaling in the text encoder (i.e. only using I-ERG), to compare against SEG (w/o extra NFE, if possible). This would prove that the non-local smoothing in ERG (L181) is clearly better than the Gaussian blurring in SEG.

Q4. In L34, the authors note that there is an oversaturation problem in CFG. How does standalone ERG tackle this problem? I did not see any experiments addressing this.

### Suggestions
S1. I believe there is a small mistake in L304, `At 512 resolution, ERG is best across all metrics, except for coverage where ERG is slightly behind CFG and SEG*`. Referring to Table 3, it seems ERG is better than SEG*, but worse than PAG*. It should probably be `...behind CFG and PAG*` in L304.

In general, I like the paper, but am concerned with some of the choices in the evaluation procedure (e.g. not using pre-trained model, converting attention UNet-based method to DiT for comparison, etc.).

If I misunderstood something in the paper or mentioned something that has already been done in the paper, please feel free to clarify my mistake.

I am willing to raise my score by 1 point if some questions are addressed.

**Ethical Concerns:**

["NO or VERY MINOR ethics concerns only"]

**Final Justification:**

My strongest concern was that the experiments in the main paper were done on "privately trained" models, despite the availability of numerous open source diffusion/flow-matching models to report results on. The authors have addressed this in the rebuttal.

Another concern was that the method ERG lacked some theoretical foundation; in the main paper, the method is presented along the lines of "we do this and it just works like CFG". The authors provide a better perspective in the rebuttal, which makes the paper stronger.

Due to these two main reasons (omitting smaller reasons for brevity), I am raising my score from Borderline Accept to Accept.

**Limitations:**

Yes, I believe the authors have addressed their limitations well.

**Paper Formatting Concerns:**

No concerns.

**Quality:**

4

**Strengths And Weaknesses:**

### Strengths
S1. The paper is well written in general. The proposed method, ERG, is well-motivated, aiming to address some of the known shortcomings of CFG.

S2. The proposed method is technically sound, built upon the idea of (a) using a weaker prediction for guidance, and (b) using the hopfield energy formulation of attention to create the weaker prediction.

S3. The authors compare their approach with several baselines for text-to-image generation, class-conditional generation, and unconditional generation. According to the results, ERG seems to simultaneously improve quality, diversity, and prompt consistency, which highlights its potential usefulness.

### Weaknesses

W1. The proposed method seems to be relevant only for a particular family of model architecture (e.g. models with attention layers). Classifier-free guidance is a very general technique that is invariant to the architecture of the generative model.

-  W1.1. Though ERG is theoretically applicable to any diffusion model with attention layers, the experiments seem to have only been performed on DiT-based architectures. It would be more convincing if the paper also covered some UNet architectures that have self-attention (e.g. EDM2).

W2. Classifier-free guidance takes a geometric weighted average between conditional and unconditional probabilities, i.e. $\hat{p}(x \mid y) \propto p(x \mid y)^{\omega}p(x)^{1 - \omega}$. It has a good probabilistic interpretation based on the score function, and can be derived from the expansion of the score function from classifier guidance. It is a bit unclear as to how an unconditional model is being guided by using a weaker unconditional model, as shown in ERG, as both models are unconditional. Is there a probabilistic interpretation of this, beyond "contrasting with weaker prediction"?

W3. It is not entirely clear what are the specific novel concepts introduced in this paper over the previous work Smoothed Energy Guidance (SEG). To my understanding, it seems that---(a) modifying the energy landscape of attention layers in diffusion models, and (b) using this phenomenon for performing a sort of guidance in conditional and even unconditional generation---have been previously established by SEG. The primary difference seems to lie in the details of how exactly the energy is modulated (temperature rescaling in ERG vs Gaussian blurring in SEG). Clearly highlighting the differences with SEG will elucidate the strengths of this paper.

---

> ### Author Rebuttal · Authors · 2025-07-30
>
> We thank the reviewer for their thoughtful feedback. We are also grateful for their acknowledgement of the motivation and technical soundness of ERG, the broad experimental work validating the method’s effectiveness across diverse settings and their positive comments on the clarity and organization of the paper.
>
> Below we address the questions raised by the review.
>
> **W1.**
> Yes, our method is based on hopfield energy interpretation of the attention so it is only applies to models with attention layers. We believe that this limitation is not major as the vast majority of large-scale state-of-the-art models currently adopt DiT-based architectures (see SD \>= 3.0, FLUX, SORA, Bagel, WAN, MovieGen etc.).
>
> Compared to some previous works, our method is less restrictive.
> For instance, Autoguidance requires access to a weaker model, and SEG/PAG were only developed for UNet models with attention, their practical usefulness for large scale SOTA models (DiT) remained unclear.
>
> **W1.1.**
> The reason for focusing on DiT architectures lies in their architecture inducing a more complex design space for methods such as SEG, and because they have become the standard for large-scale generation. However, we agree that providing evaluations on different architectures would strengthen our work. Following the setup in Section F of the appendix, we provide additional (I-)ERG results for SDXL, PixArt-alpha, DiT and EDM2 (autoguidance). SDXL and EDM2 use UNet architectures while PixArt-alpha uses local window attention. For SDXL and Pixart-alpha, we use the opensource version from Hugging Face alongside the EvalGIM evaluation framework. For EDM2 and DiT, we use the evaluation setups from their respective open-source implementations.
>
> | Model\Metric | FID | CLIP | Density | Coverage | Precision | Recall |
> | :---- | :---- | :---- | :---- | :---- | :---- | :---- |
> | PixART-alpha | 28.64 | 26.03 | 52.98 | 13.46 | 47.85 | 43.38 |
> | PixART-alpha w. ERG | 27.75 | 26.61 | 70.61 | 16.57 | 56.40 | 42.86 |
> | SDXL | 21.65 | 29.54 | 80.93 | 20.00 | 60.68 | 49.17 |
> | SDXL w. ERG | 20.94 | 29.97 | 84.05 | 20.21 | 60.66 | 54.39 |
>
> | Model \ Metric | FID | sFID | Precision | Recall | IS |
> | :---- | :---- | :---- | :---- | :---- | :---- |
> | DiT-XL/2-256 | 2.71 | 4.55 | 82.77 | 57.94 | 277.96 |
> | DiT-XL/2-256 \+ ERG |  2.36 | 4.32 | 86.9 | 60.03 | 295.03 |  |
>
> | Model \ Metric | FID | FID\_dinov2 |
> | :---- | :---- | :---- |
> | EDM2-IN-XXL-512 | 2.00 | 59.34 |
> | EDM2-IN-XXL-512 \+ ERG | 1.35 | **31.32** |
> | EDM2-IN-XXL-512-autog | **1.33** | 31.56 |
>
> **W2.**
> We thank the reviewer for this interesting question. We provide two interpretations of ERG based on entropy regularized RL and variational inference.
>
> Let $p$ be the density function for the strong model and $p^\tau$ the one induced by the attention rectification mechanism we introduce in the paper.
>
> ## I-ERG
>
> We begin with the I-ERG guidance update, which modifies the model’s score estimate as:
>
> $
> \nabla_x \log p^\tau(x) = \nabla_x \log p(x) + w \cdot \left( \nabla_x \log p(x) - \nabla_x \log p^\tau(x) \right)
> $
>
> Rewriting the right-hand side, we obtain:
>
> $
> \nabla_x \log p^\tau(x) = \nabla_x \log p(x) + w \cdot \nabla_x \log \left( \frac{p(x)}{p^\tau(x)} \right)
> $
>
> Integrating both sides with respect to $x$, we find:
>
> $
> p^\tau(x) \propto p(x) \cdot \exp \left( w \cdot R(x) \right), \quad \text{where } R(x) = \log \left( \frac{p(x)}{p^\tau(x)} \right)
> $
>
> This form reveals that I-ERG guidance parallels a posterior distribution that reweights the base model’s density $p(x)$ by an exponential function of the reward signal $R(x)$.
>
> This exponential reweighting aligns with objectives used in **entropy-regularized RL** or **KL control**, where the optimal distribution maximizes an expected reward under a KL penalty with respect to a prior:
>
> $
> \pi^* = \arg\max_{\pi}  \mathbb{E}_{x \sim \pi}[R(x)] - \frac{1}{\lambda} \mathrm{KL}(\pi \Vert  p)
> $
>
> Here, $\lambda$ serves as an inverse temperature controlling the sharpness of reward influence. The solution to this optimization problem has the form:
>
> $
> \pi(x) \propto p(x) \cdot \exp(\lambda R(x))
> $
>
> This shows that the I-ERG update implicitly implements the maximum entropy policy framework, steering the sampling distribution toward high-reward regions while maintaining proximity to the base model $p(x)$. Equivalently, using Bayes rule, it can be viewed as defining a **joint distribution** over $x$ under $p(x)$ and the energy induced by reward $R(x)$, (assuming conditional independence).
>
> ## C-ERG
>
> In the conditional setting, assume we start from a model $p(x|s)$ conditioned on some semantic signal $s$. Let $s^\tau = s + \Delta s$ represent a degraded or weakened version of the conditioning (e.g., via temperature scaling or perturbation).
>
> The C-ERG guidance update is given by:
>
> $
> \nabla_x \log p^{ERG}(x|s, s^\tau) = \nabla_x \log p(x|s) + w \cdot \left( \nabla_x \log p(x|s) - \nabla_x \log p(x|s^\tau) \right)
> $
>
> Or, equivalently:
>
> $
> \nabla_x \log p^{ERG}(x|s, s^\tau) = (1 + w) \cdot \nabla_x \log p(x|s) - w \cdot \nabla_x \log p(x|s^\tau)
> $
>
> To analyze this further, we perform a first-order Taylor expansion of the degraded score $\nabla_x \log p(x|s^\tau)$ around $s$:
>
> $
> \nabla_x \log p(x|s^\tau) \approx \nabla_x \log p(x|s) + \frac{\partial^2 \log p(x|s)}{\partial x \, \partial s^\top} \cdot \Delta s
> $
>
> Substituting this into the update:
>
> $
> \nabla_x \log p^{ERG}(x|s, s^\tau) \approx \nabla_x \log p(x|s) - w \cdot \left( \frac{\partial^2 \log p(x|s)}{\partial x \, \partial s^\top} \cdot \Delta s \right)
> $
>
> Thus, C-ERG introduces a **Jacobian-vector product correction**, nudging the sample along directions where the score is most sensitive to perturbations in the conditioning. This helps reinforce fine-grained semantic detail in generation, even when the conditioning signal is noisy or coarsened.
>
> This Jacobian-vector product has a natural interpretation: it reflects how the image-level score $\nabla_x \log p(x|s)$ changes when the conditioning $s$ is slightly perturbed. This resembles **classifier guidance**, where one steers the generation based on $\nabla_x \log p(y|x)$ — the gradient of a classifier with respect to the image.
>
> Here, the conditioning is a continuous vector $s$, and C-ERG guides generation using:
>
> $
> \frac{\partial^2 \log p(x|s)}{\partial x \, \partial s^\top} \cdot \Delta s
> $
>
> which is the **direction in image space most sensitive to changes in the semantics of $s$**. This amounts to amplifying features that are especially responsive to semantic precision.
>
> **W3.** The distinction between our method and Smoothed Energy Guidance (SEG) can be better clarified. While SEG also modifies attention energies in diffusion models for guidance—both in conditional and unconditional settings—the key differences lie in the mechanism of rectification and its broader applicability.
> Below, we summarize the main novelties of our method over SEG:
> * ERG modifies attention energies not only in the denoiser but also in the text encoder, allowing finer control over consistency and diversity.
> * Our ablations demonstrate that temperature rescaling (ERG) consistently outperforms the Gaussian blurring approach used in SEG.
> * SEG shows an unfavorable tradeoff between quality and consistency when applied to DiT architectures. We address this by introducing a kickoff threshold, which controls steerability while preserving high quality.
> * ERG is compatible with DiT-based models (and UNet models with attention) we provide clear implementation guidelines for models of varying sizes. SEG paper provides a limited scope in comparison.
> * Unlike prior works that report only limited metrics such as FID and IS, we conduct extensive evaluations across multiple generation settings. We thoroughly examine trade-offs between quality, diversity, and consistency, and offer practical insights for applying ERG across architectures.
>
> **Q1&2.** Retraining models is not required for our method. We chose to use our internal model to adhere to internal guidelines and, more importantly, to ensure tighter control over key variables that often differ across open-source models—such as training data, architecture, image resolution, task type, and the specifics of the diffusion/flow scheduler and sampler. By using an internal model, we minimize confounding factors and enable a fairer comparison of guidance methods.
> To address the reviewer’s concern, we have also added a new set of experiments using publicly available implementations of popular open-source models (see the tables above). Thus, we now cover SD 3.0, SD 3.5 (Table 8 of the appendix), SDXL, PixArt-alpha, DiT, and EDM2 (tables above).
>
> **Q3.** We believe the requested ablation is presented in Table 6 & Figure 17, and discussed in Section C.3 of the appendix. We compare I-ERG with a kickoff threshold $\kappa = 0$ with SEG, hence the only difference under this setting is the attention rectification mechanism (Gaussian blurring vs entropy rectification). Results reported showcase better detail modeling qualitatively and better scores for FID, density, coverage and CLIPScore with entropy rectification than with Gaussian blurring. If the reviewer has more specific questions about these experiments, we would be happy to address them.
>
> **Q4.** We indirectly address this in section C.2 of the appendix.
> By analyzing the orthogonal and parallel components of the guidance term, we find ERG to provide a stronger orthogonal signal wrt CFG, enabling better quality at lower guidance strengths, which reduces oversaturation issues.
> We will better clarify this link in the paper.
>
> **S1.** Thank you for pointing out this typo. It will be corrected in future versions of the paper.
>
> [1] Adjoint Matching: Fine-tuning Flow and Diffusion Generative Models with Memoryless Stochastic Optimal Control, Domingo-Enrich et al., ICLR 2025.

---

### Official Review · Reviewer_QhcC · 2025-07-03

**Clarity:** 3
**Significance:** 2
**Originality:** 2
**Rating:** 4
**Confidence:** 5

**Summary:**

This paper presents Entropy Rectifying Guidance (ERG), a guidance mechanism for diffusion and flow models designed to simultaneously improve the quality, diversity, and input consistency of generated samples, including text-to-image, class-conditional, and unconditional generation tasks. ERG operates by manipulating the attention mechanism, specifically using a Hopfield energy-inspired temperature scaling and attention modification at inference time. This yields a weaker predictive signal contrasted with the original model, thereby providing guidance without the need for an additional model or costly model evaluations. Empirical results demonstrate that ERG outperforms standard Classifier-Free Guidance (CFG) and several recent advanced baselines.

**Questions:**

Please refer to Weaknesses.

**Ethical Concerns:**

["NO or VERY MINOR ethics concerns only"]

**Final Justification:**

All my concerns have been addressed.

**Limitations:**

Yes.

**Paper Formatting Concerns:**

No.

**Quality:**

3

**Strengths And Weaknesses:**

Strengths:
1. ERG does not require model retraining, works with pre-trained transformer-based diffusion/flow models, and applies to text-to-image, class-conditional, and unconditional setups.
2. The approach extends the energy landscape of attention to invent a new, simple guidance mechanism that is broader than prior methods relying on auxiliary or weaker models.
3. This paper is well-written and well-organized. The proposed method, ERG, is easy to follow.

Weaknesses:
1. Although ERG, compared to CFG, can be applied to unconditional image generation, the performance improvement, especially in terms of FID scores, is relatively minimal.
2. The evaluation metrics used in this paper are not very standard. Commonly used metrics, such as Inception Score (IS), sFID, Precision, and Recall, are not reported, which somewhat diminishes the persuasiveness of the experimental results.
3. Other guidance techniques, such as CFG, can be applied to autoregressive image generation models. Could ERG similarly improve the performance of such models as well?

---

> ### Author Rebuttal · Authors · 2025-07-30
>
> We thank the reviewer for their thoughtful and encouraging feedback. We are particularily grateful that they appreciate the clarity and organization of the paper, the versatility and simplicity of ERG, and its ability to enhance diverse generative tasks without the need for retraining or auxiliary models.
>
> Below we address the questions raised by the review.
>
> **W1.** “*Performance improvement is relatively minimal, especially in terms of FID*”
>
> We acknowledge the reviewer’s observation. Our evaluation setup is designed to optimize **overall generation quality**, balancing *fidelity, diversity, and consistency*—not just a single metric like FID. This makes direct comparisons on FID alone potentially misleading.
>
> As shown by [3], FID-optimal checkpoints often lie far from the Pareto front when jointly considering realism, consistency, and diversity, and in models such as SDXL, optimizing for FID can significantly reduce realism (Section 4.1, Fig. 3).
> Dhariwal et al. [4] further show that lower guidance scales minimize FID and recall, while higher guidance scales improve perceptual sharpness and structure, measured by IS and precision even though they increase FID (Figure 3).
>
> However, to directly address this concern, we conducted an additional experiment where we **tuned each method specifically to optimize FID**, selecting the best hyperparameters (e.g., guidance scale, ERG scale) for each.
>
> | Method | CFG | ERG | APG | ERG+APG | PAG | SAG | SEG | CADS | ERG+CADS | AG |
> | :---- | :---- | :---- | :---- | :---- | :---- | :---- | :---- | :---- | :---- | :---- |
> | FID | 5.25 | **4.93** | 6.62 | 5.24 | 7.00 | 5.28 | 6.72 | 5.12 | 5.00 | 7.11 |
>
> CFG achieves its best FID at a guidance scale of **1.25**, while ERG achieves its optimal FID of **4.93** using a guidance scale of **1.25** and an ERG scale of **2.0**. This demonstrates that ERG is capable of outperforming prior guidance methods (e.g., SEG, APG) when optimized for FID alone, in addition to providing better trade-offs under broader evaluation criteria as can be seen in the Pareto fronts in Figure 2 of the main paper.
>
> **W2.** “*The evaluation metrics used in this paper are not very standard. Commonly used metrics, such as Inception Score (IS), sFID, Precision, and Recall, are not reported, which somewhat diminishes the persuasiveness of the experimental results.*”
>
> We thank the reviewer for their insight on this point. We originally chose to report a subset of metrics that robustly capture the distinct facets of generative performance without overwhelming the reader. Specifically, we believe that density and coverage (D\&C) offer more reliable and interpretable fidelity and diversity evaluations than precision and recall (P\&R), as detailed in \[1\]. While P\&R have advanced the separation of fidelity and diversity, they exhibit key limitations:
>
> * They fail to detect matches between identical distributions, often reporting suboptimal values even when real and generated data are sampled from the same distribution (\[1\] \- Fig. 3).
> * They are vulnerable to outliers, as a single real or fake outlier can inflate the manifold estimation and artificially boost scores (\[1\] \- Fig. 3, left).
> * They depend on arbitrarily chosen hyperparameters, such as the number of nearest neighbors k, without principled guidance for selection ([1] - sec. 3.1).
> * In contrast, D\&C address these issues by using a superposition-based manifold estimation that is more robust to data sparsity and outliers, yield analytically tractable behavior under ideal conditions ([1] - sec. 3.3), and support systematic hyperparameter tuning ([1] - sec. 3.4).
>
> Therefore, we opted for D\&C as they faithfully reflect the fidelity-diversity trade-offs of generative models in both synthetic and real-world scenarios.
>
> That said, we fully agree on the value of standard benchmarks for comparability. We now include Inception Score (IS), sFID, and Precision & Recall for completeness.
>
> | Method | CFG | APG | CADS | PAG | SAG | SEG | AutoGuidance | ERG | ERG+APG | ERG+CADS |
> | :---- | :---- | :---- | :---- | :---- | :---- | :---- | :---- | :---- | :---- | :---- |
> | Precision | 65.71 | 66.34 | 66.42 | 66.42 | 64.97 | 61.22 | 61.48 | 70.92 | 69.50 | **72.72** |
> | Recall | 43.95 | 45.15 | 45.75 | 43.94 | 47.98 | 36.88 | 34.60 | 41.43 | **50.25** | 38.89 |
> | sFID | 26.53 | 20.31 | 18.20 | 27.21 | 28.55 | 33.80 | 14.75 | 19.56 | **9.30** | 15.32 |
> | IS | 37.25 | 43.76 | 40.75 | 40.50 | 38.28 | 40.64 | 41.81 | 43.16 | **53.47** | 42.52 |
>
> For precision & recall, we observe that similar trends to density and coverage with ERG+CADS achieving the best precision (72.72). ERG on its own achieves a precision of 70.92 compared to 65.71 for CFG.
> Recall is slightly lower when using ERG alone (41.43 vs 43.95 for CFG), but ERG+APG achieves the best recall of 50.25.
> For Inception Score, ERG improves over the CFG baseline (43.16 vs 37.25 for CFG) and achieves the best score when combined with APG (53.47).
> Similarly, ERG+APG results in significant sFID and Inception score improvement, with ERG alone on par with APG.
>
> **W3.** “*Other guidance techniques, such as CFG, can be applied to autoregressive image generation models. Could ERG similarly improve the performance of such models as well?*”
>
> We appreciate the reviewer's suggestion. While guidance methods like CFG have been successfully applied to autoregressive image generation, extending ERG to this setting is non-trivial and would require further research.
> In diffusion and flow-based models, such guidance can be interpreted theoretically due to the iterative nature of the sampling process which operates on the global image representation.
> In contrast, models like LlaMAgen \[2\] sequentially predict the series of $h \times w$ tokens without access to the full image context, no refinement process similar to diffusion/flow sampling occurs.
> We conducted a preliminary investigation using the LLaMAgen model but did not observe stable improvements with an out-of-the-box implementation of ERG compared to CFG.
>
> This suggests that directly transferring ERG to autoregressive architectures may not be straightforward, potentially due to differences in the way attention and generation unfold across time steps in such models, which renders some parameters in ERG incompatible, such as the I-ERG kickoff threshold which does not have a direct parallel in the autoregressive setting.
>
> Nonetheless, we believe this is a promising direction for future work and are interested in exploring how ERG-like mechanisms could be adapted to autoregressive settings more effectively.
>
> \[1\] Reliable fidelity and diversity metrics for generative models. Naeem et al., ICML, 2020.
>
> \[2\] Autoregressive Model Beats Diffusion: Llama for Scalable Image Generation, Sun et al., 2024.
>
> \[3\] Consistency-diversity-realism Pareto fronts of conditional image generative models, Astolfi et al., NeurIPS, 2024.
>
> \[4\] Diffusion Models Beat GANs on Image Synthesis, Dhariwal et al., NeurIPS, 2021.

---

> > ### Comment · Reviewer_QhcC · 2025-08-05
> >
> > Thank you for your rebuttal. All my concerns have been addressed.

---

### Decision · Program_Chairs · 2025-09-17

**Decision:**

Accept (poster)

**Comment:**

This paper presents Entropy Rectifying Guidance (ERG), a guidance mechanism for diffusion and flow models designed to simultaneously improve the quality, diversity, and input consistency of generated samples, including text-to-image, class-conditional, and unconditional generation tasks. ERG operates by manipulating the attention mechanism, specifically using a Hopfield energy-inspired temperature scaling and attention modification at inference time. This yields a weaker predictive signal contrasted with the original model, thereby providing guidance without the need for an additional model or costly model evaluations. Empirical results demonstrate that ERG outperforms standard Classifier-Free Guidance (CFG) and several recent advanced baselines.

The reviewers liked the creativity, novelty, simplicity, and soundness of the method. They raised various concerns, which the authors addressed successfully during the discussion period. In the end, all reviewers endorsed the acceptance of the paper.